# Computed Tomography Utilization in the Management of Children with Mild Head Trauma

**DOI:** 10.3390/children10071274

**Published:** 2023-07-24

**Authors:** Ernest Leva, Minh-Tu Do, Rachael Grieco, Anna Petrova

**Affiliations:** Department of Pediatric, Rutgers Robert Wood Johnson Medical School, New Brunswick, NJ 08903-0019, USA; levaeg@rwjms.rutgers.edu (E.L.); domi@rwjms.rutgers.edu (M.-T.D.); griecora@rwjms.rutgers.edu (R.G.)

**Keywords:** children, computer tomography, mild head trauma

## Abstract

This study demonstrates the trend of computed tomography (CT) usage for children with mild traumatic brain injury (mTBI) in the context of the initiation of the *Safe CT Imaging Collaborative Initiative* to promote the Pediatric Emergency Care Applied Research Network (PECARN) rules at the acute care hospitals in New Jersey. We used administrative databases of 10 children’s and 59 general hospitals to compare CT rates before 2014–2015, during 2016, and after the initiation of the program (2017–2019). The CT usage rates at baseline and the end of surveillance in children’s hospitals (19.2% and 14.2%) were lower than in general hospitals (36.7% and 21.0%), *p* < 0.0001. The absolute mean difference from baseline to the end of surveillance in children’s hospitals was 5.1% compared to a high of 9.7% in general hospitals, medium-high with 13.2%, and 14.0% in a medium volume of pediatric patients (*p* < 0.001–0.0001). The time-series model demonstrates a positive trend of CT reduction in pediatric patients with mTBI within four years of the program’s implementation (*p* < 0.03–0.001). The primary CT reduction was recorded during the year of program implementation. Regression analysis revealed the significant role of a baseline CT usage rate in predicting the level of CT reduction independent of the volume of pediatric patients and type of hospital.

## 1. Introduction

According to the Centers for Disease Control and Prevention (CDC), children have one of the higher rates of traumatic brain injury (TBI) related to emergency department (ED) visits [1]. In addition to the risk of death and severe disability, exposure to cranial computed tomography (CT), which is a primary diagnostic tool for managing TBI in childhood, could result in a lifelong threat of malignancy [2,3,4]. At the same time, CT is characterized by low sensitivity and insufficient capacity to diagnose diffuse brain damage [5] and predict post-traumatic outcomes [6]. Although CT is not recommended for the routine evaluation of pediatric blunt head trauma [7], which is generally mild and not clinically consequential [8,9,10,11], the head has been recognized as the most common CT-imaged region for children of any age [3]. Extensive national data showed an increase in CT usage up to the year 2005 in the management of pediatric patients with mild head trauma [12] without any decrease in the subsequent five [13] and ten [14] years. Several algorithms were developed [15,16] to assist physicians in avoiding unnecessary diagnostic imaging in children with mild TBI (mTBI). One of them is Pediatric Emergency Care Applied Research Network (PECARN) rules recommended by the CDC [7]. According to the CDC, if PECARN rules are utilized appropriately, up to 25% of cranial CTs performed in the ED setting could be safely avoided [10]. A survey of healthcare professionals revealed the predominant use of the PECARN rules to manage pediatric patients presenting to the ED with mTBI [17]. Even though nationwide ED data showed no reduction in the CT rates after the publication of the PECARN rules [18], implementation of such an algorithm in single ED settings allowed the achievement of a significant decrease in the use of CT scans in children with mTBI [19,20,21,22]. The American College of Radiology also supports using PECARN criteria to identify very low-risk pediatric patients with mTBI who can safely avoid CT imaging [23].

In 2016, the New Jersey Hospital Association (NJHA), a not-for-profit trade organization that supports quality-improvement programs and services in New Jersey hospitals [24], introduced the *Safe CT Imaging Collaborative Initiative* to promote PECARN rules usage in EDs of hospitals providing acute care for the pediatric population. To our knowledge, no study has evaluated the effort of professional organizations in promoting PECARN rules in hospital settings to reduce the use of CT in managing pediatric patients with mTBI. We developed this study to examine whether NJHA initiatives were associated with changes in CT utilization for pediatric patients with mTBI in the EDs of acute care hospitals in New Jersey. Our study’s findings could be valuable for justifying the need to enhance efforts to implement medical evidence in emergency care for pediatric patients.

## 2. Materials and Methods 

This study analyzed administrative ED data from 72 acute care hospitals participating in the *Safe CT Imaging Collaborative* Initiative. The primary purpose was to identify the overall trend of the CT rate from before 2014–2015 and three years after 2016, the year of implementation of the *Safe CT Imaging Collaborative* for all affiliated children and general hospitals and the pediatric ED visit-level rates.

### 2.1. Description of Safe CT Imaging Collaborative Initiative 

NJHA developed *Safe CT Imaging Collaborative* to promote adherence to PECARN rules for managing pediatric patients with head trauma in the EDs of affiliated acute care hospitals in New Jersey to reduce the use of CT for children with mTBI. The NJHA reached out to hospitals and contacted medical and nursing officials to participate in the *Safe CT Imaging Collaborative Initiatives*. In 2016, the NJHA, in collaboration with the Council of the Children’s Hospitals, disseminated material containing evidence of PECARN rules in reducing CT usage in managing mTBI in children seen at EDs. Moreover, NJHA organized hospital-wide educational sessions, ground rounds, and lectures for physicians and other healthcare professionals in the EDs of the participating hospitals.

### 2.2. Characteristics of Collected Data 

The hospitals engaged in the program provided administrative ED data for each year before 2014–2015, during 2016, and after 2017–2019, implementing the *Safe CT Imaging Collaborative Initiative*. Hospital-based aggregative data defined yearly included the number of pediatric ED patients overall and specifically with mTBI with and without the use of a cranial CT for each demographic category: age (0–2, 3–5, 6–10, 11–14, and 15–17 years); sex (male/female); race (White, Black, Asian, American Indian, Pacific Islander, multiracial, and other); ethnicity (Hispanic/Non-Hispanic); and type of insurance (Medicare, Medicaid, private insurance, and uninsured). The mTBI included head injury without further description, concussion without loss of consciousness, mild traumatic brain injury, mild head injury, and minor head trauma coded by the ICD-9 (850–854, 920, 910, 925, 959.01) and ICD-10 (S00.0, S00.1, S00.2, S00.8, S00.9, S09.8, and S09.9). The current procedural terminology (CPT) codes for head CT without (70,450) and with (70,460) contrast were used to identify CT usage for pediatric patients seen in the ED with head trauma. 

### 2.3. Hospital Type and ED Classification 

We used the New Jersey Department of Health and Human Services hospitals’ designation and pediatric ED visit-level rates [25,26] to identify the children’s and general hospitals and characterize the EDs with a low (<1800), medium (1800–4999), medium-high (5000–9999), and high (≥10,000) annual volume of pediatric patients. 

### 2.4. Data Presentation 

The final analysis included hospitals that submitted data for the base years (2014–2015), during 2016, and at least two years out of three (2017–2019) after the implementation of the *Safe CT Imaging Collaborative initiative*. We stabilized the baseline CT rate by estimating the average of 2014 and 2015 data for further time-trend analysis of equal-year intervals. We calculated the proportion (%) of pediatric patients with mTBI and CT use overall and by age categories, gender, race/ethnicity, and kind of health insurance coverage. Patients of American Indian and Pacific Islander and multiracial backgrounds were placed in the “other” racial group, and those covered by Medicare were not included in the final analysis because of an insufficient annual quantity. We analyzed the proportion of mTBI among ED pediatric patients and CT use by type of hospitals (children’s vs. general) and pediatric ED visit-level rates (high, medium-high, medium, and low). 

### 2.5. Statistical Analysis

We used Chi-square statistics and ANOVA followed by a post hoc Tukey honest significant difference test to compare the categorical and continuous variables. The significance of the annual change in CT usage rate was analyzed using a t-test for dependent samples to determine the absolute change (AC) and absolute percentage change (APC) yearly [27]. Such analysis was conducted for each hospital from the baseline (2014–2015) through 2016–2019. The comparison included the following periods: baseline (2014/2015)–2016, 2016–2017, 2017–2018, 2018–2019, and 2014/2015–2019. CT usage rate and APC trends were presented in total and for the hospital designation (children’s vs. general) and pediatric ED visit-level rates. We also categorized the alteration of CT rates from the baseline to the end of surveillance for each hospital as no change, increase, or decrease if the CT rate difference from baseline to the end of the surveillance period in 2019 was equal to 0, more than 0, or less than 0. We presented trends using a time series model that allows us to predict the alteration of CT usage rate at specific time values for each type of hospital.

The findings are presented as mean (%) and difference in means (diff. mean) of mTBI and CT rates and included the corresponding 95% confidence interval (95% CI), overall and specifically for each demographic category. We constructed a stepwise linear-regression model to identify the factors that predicted the absolute change of CT usage rate from baseline, including baseline CT rate, the annual number of pediatric ED patients, and the type of hospital (children’s vs. general). All statistical tests were 2-sided and presented with the significance level set at a *p*-value of <0.05. Study results were analyzed using STATISTICA 13.2 (StatSoft, Tulsa, OK, USA).

## 3. Results 

Of 72 hospitals engaged in the *Safe CT Imaging Collaborative Initiative*, 69 hospitals (10 children’s and 59 general) completed the submission of the required administrative data. Overall, 4,086,247 pediatric patients visited the EDs in 69 hospitals during the six years of surveillance. Table 1 characterizes EDs by the type of hospital (children’s vs. general) and volume of pediatric visits. All EDs in the children’s hospitals had ranked as having a high volume of ED visits. Among the EDs located in the general hospitals, 19 (32.2%) had ranked with high, 23 (39.0%) with medium-high, 15 (25.4%) with medium, and 2 (3.4%) with a low volume of pediatric visits. As shown in Table 1, the mean annual number of pediatric visits at 10 EDs in the children’s hospitals was higher than in the 19 EDs located at general hospitals with high-volume pediatric patients (*p* < 0.001). We excluded two hospitals with a low (less than 1000) annual volume of pediatric ED visits.

### 3.1. Proportion of Children with mTBI at EDs during Surveillance

During the study period, mTBI constituted 3.2% (95% CI 3.0%, 3.3%) of pediatric ED visits, including 2.7% (95% CI 2.3%, 3.1%) in children’s and 3.3% (95% CI 3.1%, 3.4%) in general hospitals (*p* < 0.02). The average rates of pediatric patients seen with mTBI at the EDs of general hospitals with medium (3.4%, 95% CI 3.1%, 3.7%), medium-high (3.3%, 95% CI 3.1%, 3.4%), and high (3.0%, 95% CI 2.7%, 3.3%) volumes of pediatric patients were comparable (*p* = 0.09). The mTBI rates during the baseline period (2014–2015) were similar overall and based on the patient’s characteristics (age, sex, race, and ethnicity) and insurance coverage (Table 2). Data presented in Table 2 showed a significant reduction of the mTBI rates in 2016 except for children 0 to 5 years old, other races, and Hispanic ethnicity. The variability of mTBI rates from 2017 to 2019 did not reach statistical significance.

### 3.2. CT Use in Children with mTBI during Surveillance 

We determined that CT usage during the surveillance period from baseline to 2019 had reduced in 61 (91.0%) hospitals by an average of −16.5% (95% CI-13.6%, -19.4%) and increased in 6 (9.0%) hospitals by an average of 2.9% (95% CI 0.65%, 5.1%). CT use increased in two children’s and four general hospitals with medium-high (*n* = 3) and medium (*n* = 1) volumes of pediatric ED visits. Table 3 shows comparable CT usage rates before initiating the *Safe CT Imaging Collaborative Initiative* (2014 and 2015), irrespective of sex, age, race, ethnicity, and health insurance type. CT use significantly reduced in 2016, the year of implementation of the *Safe CT Imaging Collaborative Initiative,* for almost all categories of patients, and in 2017 for patients aged 0–1 and 15–17 years and Asian and other races. 

### 3.3. CT Use Based on Type of Hospital and Pediatric Patient Values 

Before implementing the *Safe CT Imaging Collaborative* (2014–2015), CT was used to manage 34.7% (95% CI 31.1%, 38.3%) of children with mTBI. The baseline CT rate in children’s hospitals (19.2%, 95% CI 13.3%, 25.1%) was significantly lower than in general hospitals overall (36.7%, 95% CI 33.0, 40.5) and with high (34.2%, 95% CI 27.3%, 41.1%), medium-high (40.5%, 95% CI 34.9, 46.0%), and medium (37.7%, 95% CI 29.6% 45.9%) ED volume of pediatric patients (*p* < 0.0001). The baseline CT rates in general hospitals with different pediatric ED visit rates were comparable (*p* = 0.41–0.95). The CT rate at the end of surveillance (2019) in the children’s hospitals was lower than in general hospitals (14.1%, 95% CI 11.1%, 17.1% vs. 21.0%, 95%CI18.3%, 23.7%, *p* < 0.001). The difference in CT usage rate between the children’s hospitals and the general hospitals with high (18.2%, 95% CI 13.2%, 23.3%) and medium (18.7%, 95% CI 14.1%, 23.3%) pediatric patient ED visit rates did not reach statistical significance (*p* = 0. 71–0.55). However, it was lower than in general hospitals with medium-high pediatric ED visit rates (24.9%, 95% CI 20.6%, 29.2%, *p* < 0.02). The absolute mean difference of CT usage reduction from the baseline to 2019 in children’s hospitals was lower (5.1%, 95% CI 0.69, 9.5) than in general hospitals with high (9.7%, 95% CI 11.2%, 19.8%), medium-high (13.2%, 95% CI 9.7%, 21.4%), and medium (14.0%, 95%CI11.0%, 28.0%) volume of pediatric ED patients (*p* < 0.01–0.0001). A linear-regression model revealed the association of the absolute size of CT usage change with the baseline CT rate (β = 0.758, 95% CI 0.571, 0.946, *p* < 0.00001), but not with the number of pediatric patients and type of hospital (*p* = 0.88–0.96).

### 3.4. CT Trend Analysis from Baseline (2014/2015) to the End of Surveillance (2019)

Figure 1 illustrates data from a time series model identifying a significant reduction of absolute CT usage rates from baseline to the end of surveillance in the children’s and general hospitals with different ED pediatric visit rates. Over time, CT reduction can be calculated using the equations where x is a point of observation for children’s hospitals (18.85–1.23*x), general hospitals with high volume (34.15–3.43*x), medium-high volume (41.47–3.81*x), and medium volume (37.93–3.9*x) of pediatric patients. 

### 3.5. Annual APC of CT Use after Implementing the Safe CT Imaging Collaborative Initiative

Table 4 presents the annual APC of CT use after implementing the *Safe CT Imaging Collaborative Initiative*. The level of CT reduction decreased from 2016 to 2017 and from 2018 to 2019. The APC of CT usage rates significantly decreased in 2016 in general hospitals and from 2016 to 2017 in general hospitals with medium-high ED visits. The reduction of APC of CT-usage in children’s hospitals decreased at the end of the surveillance.

## 4. Discussion 

The results of our study demonstrate a significant reduction in the CT utilization rate in pediatric patients with mTBI who were seen in 91% of the 69 New Jersey hospitals engaged in the *Safe CT Imaging Collaborative Initiative*. CT usage rates decreased by nearly 15% (*p* < 0.001) from the baseline during the four years of surveillance, predominantly during the year of program implementation followed by an average annual reduction of 1.9% during the three years of follow-up. Regression models, controlled for the type of hospital (children’s vs. general) and the yearly volume of pediatric ED patients revealed a significant role of the baseline CT rate in reducing CT usage. Several short quality improvements (QI) projects that primarily represented single hospital settings showed a significant reduction in CT usage rates after the implementation of the PECARN rules from 26.7% (2010) to 18.9% (2011/2012) [28], from 41.8% (2012) to 27.7% (2013/2014) [29], from 29.2% (2013/2014) to 17.4% (2014/2015) [20], and from 37.7% (2014/2015) to 16.9% (2016/2017) [30]. Such studies also illustrate the relationship between the reduction in CT usage and baseline CT rates. Moreover, the reduction in CT usage was more significant for the general hospitals with initially higher CT rates [20,28,29,30] than in the children’s hospitals [19,22], similar to our findings. However, the CT utilization rate in the children’s hospitals in New Jersey was higher than the 6.3% recorded during the five years from 2012 to 2016 in the Children’s Hospital of Philadelphia [22]. A multifaceted QI program that included implementing PECARN rules in Boston Children’s Hospital resulted in a one-year CT reduction from 21% to 15% and 9% after the inclusion of individual provider feedback in 2012 [19]. Implementing the PECARN rules in the general hospitals affiliated with Boston Children’s Hospital also demonstrated a reduction in CT usage that did not reach that in the parent hospital [19]. Correspondingly, we revealed that despite a substantially higher reduction of CT usage for pediatric mTBI, EDs located in general hospitals had CT rates higher than those in the children’s hospitals (21.0%, 95% CI 18.3%, 23.7% vs. 14.1%, 95% CI 11.1%, 17.1%, *p* < 0.001). We understand that other hospital- and patient-based factors could also be associated with the utilization of CT in pediatric mTBI. It has been shown that CT use is less likely in pediatric patients presenting to the EDs located in non-trauma centers [11,29] and non-teaching hospitals [13,14,18]. Studies showed that pediatric emergency medicine- (PEM) trained physicians are less likely to use CT in managing mTBI [22] even though individual variability in CT use among PEM-trained and general pediatricians has been reported [31]. Moreover, the pediatricians’ experience [20] and risk tolerance [32], as well as parental imaging expectations [20], but not parental involvement in the decision to use CT [33,34], may affect the practice of cranial CT scanning. The few reports that analyzed the CT use rates based on the demographic characteristics of mTBI cases showed an increased [14,35] or no impact [28] of age and race/ethnicity [35]. We recorded a favorable trend of CT performance in managing children with mTBI of all ages, sex, races/ethnicity, and health insurance status.

We recognized the limitation of evaluating the value of *the Safe CT Imaging Collaborative Initiative* in CT usage reduction because of the lack of data regarding the initiation, maintenance, and adherence of PECARN rules in managing mTBI in pediatric patients at each of the ED settings before and after initiation of the planned campaign. This study analyzed CT rates by the type of hospital and the annual volume of pediatric ED patients, but other factors could also influence the CT usage rate in children with mTBI who are seen in the EDs. Another limitation is the lack of knowledge regarding the ED physicians’ training that could impact CT use in pediatric head trauma. Besides, the collected data did not provide information about the safety of the reduction in CT use, such as misdiagnosis of the severity of the injury, which could have been an essential addition to our findings. However, this study utilized administrative ED data that *Academic Emergency Medicine* identified as suitable for large-scale trends analysis of emergency imaging research [36]. The data used is less likely to have a selection bias [37] because it represented EDs providing acute care for the entire pediatric population of New Jersey. Moreover, gathering the variables of interest was standardized to collect data systematically and uniformly from each hospital from the base year through the surveillance period.

## 5. Conclusions

The *Safe CT Imaging Collaborative Initiative* campaign was associated with a positive trend toward reducing the use of CT for evaluating children with mTBI seen in the EDs of acute care hospitals in New Jersey. The primary CT reduction was recorded during the year of program implementation and was also significantly predicted by the baseline CT usage rates. Further efforts will be needed to strengthen the utilization of PECARN rules to eliminate the inequality in practicing CT in pediatric patients with mTBI in the ED setting.

## Figures and Tables

**Figure 1 children-10-01274-f001:**
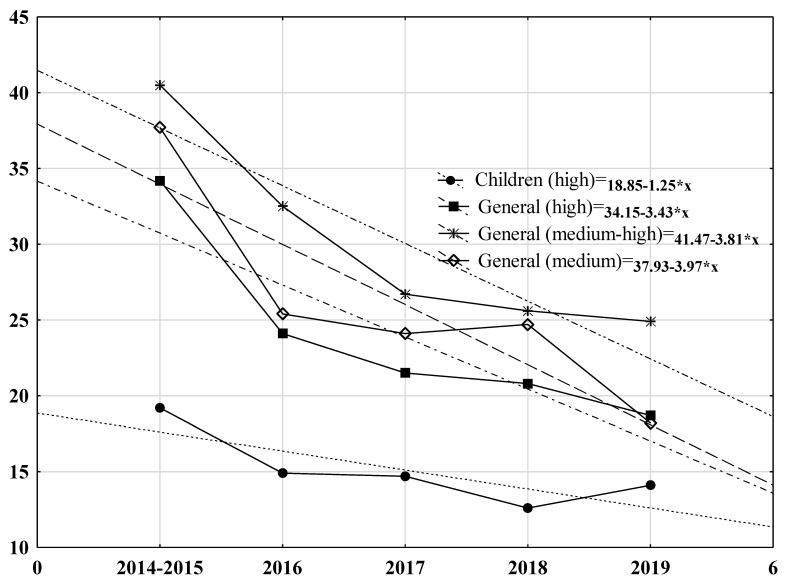
Trend of CT rate reduction during surveillance period in children’s (*p* < 0.03) and general hospitals (*p* < 0.001). Legend: Time series model presents the equation for each type of hospitals. Trendline (linear) presented as dash.

**Table 1 children-10-01274-t001:** Pediatric EDs classified by the type of hospital and volume of pediatric visits *.

Type of Hospital and ED Volume	Year (2014–2019)					
	2014	2015	2016	2017	2018	2019
**Children** **General** high	**20,946** 15,707–26,186 **14,317** 12,579–16,054 **‡**	**21,115** 15,597–26,634 **143,29** 12,536–16,121 **‡**	**21,766** 15,849–27,683 **14,341** 12,493–16,188 **‡**	**21,766** 15,849–27,683 **14,341** 12,493–16,189 **‡**	**21,696** 16,364–27,027 **14,505** 12,465–16,544 **‡**	**21,310** 16,243–26,377 **15,156** 12,974–17,337 **‡**
**General** Medium- high Medium	**7292** 6561–8023 **3315** 2800–3829	**7221** 6595–7848 **3560** 3080–4040	**7389** 6790–7988 **3560** 3090–4030	**7586** 7006–8165 3497 3050–3944	**7524** 6960–8088 3359 2905–3813	**7416** 6940–8093 3644 3191–4096

* Data presented as mean (95%CI); **‡**
*p* < 0.0001 presents comparison of annual number of pediatric Emergency Department (ED) visits between high volume EDs located in children’s and general hospitals.

**Table 2 children-10-01274-t002:** mTBI rates in pediatric patients seen at EDs (2014–2019) *.

Characteristics	2014	2015	2016	2017	2018	2019
**Overall mTBI rate**	4.0 (3.5–4.4)	3.8 (3.4–4.3)	2.9 **‡** (2.6–3.2)	2.7 (2.5–3.2)	2.7 (2.4–3.0)	2.8 (2.5–3.1)
**Sex**						
Male	4.5 (4.0–5.0)	4.3 (3.8–4.8)	3.2 **‡** (2.9–3.6)	3.3 (2.9–3.7)	3.0 (2.6–3.3)	3.1 (2.8–3.5)
Female	3.4 (3.0–3.8)	3.3 (2.9–3.7)	2.5 **‡** (2.2–2.8)	2.4 (2.1–2.7)	2.3 (2.0–2.6)	2.7 (2.2–2.8)
**Age (years)**						
0–1	4.4 (3.5–5.4)	4.0 (3.5–4.7)	3.7 (3.2–4.2)	3.7 (3.2–4.2)	3.5 (3.1–3.9)	3.8 (3.3–4.4)
2–5	3.6 (3.0–4.0)	3.7 (3.1–4.3)	3.1 (2.7–3.5)	3.1 (2.7–3.5)	2.9 (2.5–3.3)	3.1 (2.6–3.6)
6–10	3.5 (3.0–3.9)	3.3 (2.9–3.8)	2.5 **‡** (2.2–2.8)	2.5 (2.2–2.8)	2.3 (2.0–2.6)	2.5 (2.2–2.8)
11–14	4.6 (4.1–5.1)	4.3 (3.8–4.8)	2.8 **‡** (2.5–3.1)	2.7 (2.4–3.1)	2.5 (2.2–2.9)	2.7 (2.4–3.0)
15–17	4.6 (4.0–5.1)	4.4 (3.9–4.9)	2.7 **‡** (2.4–3.0)	2.4 (2.1–2.7)	2.4 (2.1–2.7)	2.4 (2.1–2.7)
**Race**						
White	4.9 (4.3–5.4)	4.7 (4.2–5.2)	3.4 **‡** (3.0–3.8)	3.3 (3.0–3.9)	3.3 (2.9–3.6)	3.6 (3.5–3.9)
Black	3.0 (2.7–3.4)	3.1 (2.7–3.4)	2.3 **‡** (2.1–2.6)	2.3 (2.0–2.6)	2.0 (1.8–2.2)	2.3 (2.1–2.8)
Asian	4.0 (3.3–4.8)	3.5 (2.8–4.1)	2.7 **‡** (2.2–3.2)	2.8 (2.2–3.3)	2.9 (2.1–3.6)	2.5 (2.0–3.0)
Other	3.0 (2.5–3.4)	2.9 (2.6–3.3)	2.6 (1.8–3.5)	2.9 (1.9–2.6)	2.3 (1.9–2.6)	2.6 (2.0–3.1)
**Ethnicity**						
Hispanic	2.6 (2.3–2.9)	2.7 (2.4–3.0)	2.2 (2.0–2.5)	2.2 (1.9–2.5)	1.9 (1.7–2.2)	2.1 (1.8–2.3)
Non–Hispanic	4.3 (3.8–4.8)	4.2 (3.7–4.6)	3.1 **‡** (2.8–3.4)	3.1 (2.7–3.4)	2.9 (2.6–3.3)	3.1 (2.7–3.4)
**Insurance**						
Medicaid	2.8 (2.4–3.2)	2.5 (2.2–2.8)	2.1 (1.9–2.4)	2.1 (1.8–2.4)	1.9 (1.7–2.2)	2.1 (1.9–2.4)
Commercial	4.9 (4.3–5.4)	4.8 (4.3–5.3)	3.5 **‡** (3.1–3.9)	3.5 (3.1–3.9)	3.3 (2.9–3.7)	3.5 (3.1–3.9)
Uninsured	3.4 (3.0–3.9)	3.3 (2.8–3.8)	2.4 **‡** (2.0–2.7)	2.6 (2.1–3.0)	2.9 (2.0–3.8)	2.2 (2.1–2.8)

* Data presented as mean rate (%) with 95%CI; **‡**
*p* < 0.01–0.0001 presents year when the rate of pediatric patients with mTBI reduced at EDs.

**Table 3 children-10-01274-t003:** CT rates (%) in pediatric ED patients with mTBI (2014–2019) *.

Factors	2014	2015	2016	2017	2018	2019
**Overall** **CT rate**	35.7 (32.1–39.4)	33.7 (30.0–37.4)	25.7 **‡** (22.8–28.6)	22.2 (19.7–24.8)	21.9 (19.2–24.7)	20.0 (17.6–22.3)
**Sex**						
Male	35.8 (32.2–39.5)	34.4 (30.5–38.2)	26.2 **‡** (23.2–29.1)	21.7 (18.9–24.5)	21.9 (19.0–24.7)	20.3 (17.8–22.7)
Female	35.6 (31.8–39.4)	32.7 (28.9–36.40	25.0 **‡** (21.8–28.1)	23.1 (20.4–25.8)	22.0 (19.1–24.9)	19.5 (16.9–22.1)
**Age (years)**						
0–1	21.4 (18.0–24.9)	21.9 (17.9–25.8)	16.9 (13.9–19.9)	14.3 **‡** (11.2–17.4)	14.4 (12.0–16.7)	12.3 (9.9–14.7)
2–5	22.2 (18.6–25.8)	21.6 (17.9–25.3)	15.1 **‡** (12.7–17.5)	13.4 (10.9–15.9)	13.2 (10.8–15.6)	12.6 (10.7–14.6)
6–10	33.4 (28.9–37.9)	30.2 (26.3–34.1)	22.0 **‡** (18.8–25.0)	18.1 (15.0–21.2)	17.5 (14.2–20.8)	14.2 (11.6–16.8)
11–14	45.8 (41.5–50.1)	41.4 (37.1–45.7)	33.2 **‡** (28.6–37.9)	29.0 (25.2–32.7)	28.8 (24.7–32.9)	28.7 (24.9–32.5)
15–17	54.9 (50.5–59.4)	52.1 (47.3–56.9)	46.9 (42.0–51.7)	43.6 **‡** (39.4–47.7)	41.6 (36.7–46.6)	37.2 (33.1–41.3)
**Race**						
White	37.6 (33.7–41.4)	36.0 (32.1–40.0)	26.9 **‡** (23.5–30.2)	23.2 (20.3–26.1)	22.2 (19.3–25.4)	21.8 (18.8–24.7)
Black	35.0 (30.3–39.6)	33.3 (29.0–37.6)	25.0 **‡** (21.2–28.8)	22.5 (18.8–26.3)	20.5 (16.5–24.5)	19.2 (15.8–22.5)
Asian	33.8 (26.2–41.4)	30.4 (23.4–37.3)	34.3 (25.2–43.2)	23.4 **‡** (15.6–31.1)	19.2 (12.4–25.9)	20.7 (13.8–27.6)
Other	34.3 (28.8–39.8)	30.0 (25.7–34.4)	26.8 (22.4–31.2)	20.6 **‡** (22.4–31.2)	21.1 (16.8–25.4)	21.9 (17.7–26.1)
**Ethnicity**						
Hispanic	35.3 (30.3–40.4)	30.7 (26.4–35.1)	23.6 **‡** (19.1–28.1)	20.8 (17.5–24.1)	20.0 (16.8–23.3)	20.5 (17.1–23.9)
Non–Hispanic	36.1 (32.4–39.8)	33.8 (30.1–37.6)	26.2 **‡** (23.2–29.1)	22.4 (19.7–25.0)	22.4 (19.5–25.3)	20.4 (17.8–23.1)
**Insurance**						
Medicaid	36.2 (31.4–41.1)	31.8 (27.7–35.9)	21.8 **‡** (18.4–25.1)	17.4 (14.6–20.2)	20.2 (16.4–23.9)	18.1 (15.0–21.2)
Commercial	36.5 (32.8–40.2)	34.8 (30.9–38.7)	27.2 **‡** (24.0–30.4)	23.7 (20.7–26.7)	22.3 19.2–25.4	21.6 (18.7–24.6)
Uninsured	37.9 (32.2–43.6)	33.0 (27.6–38.5)	26.4 **‡** (20.8–31.9)	22.8 (17.8–27.9)	22.1 (16.9–27.4)	17.8 (14.2–21.3)

* Data presented as mean rate (%, with 95%CI)**; ‡**
*p* < 0.01–0.0001 presents year when the rate of CT used in pediatric patients with mTBI.

**Table 4 children-10-01274-t004:** APC/year based on type of hospital and annual volume of ED pediatric visits *.

Hospital/ED Volume	2014/2015–2016	2016–2017	2017–2018	2018–2019	2014/2015–2019
**Total**	−9.0 (−6.5, −11.6) ***	−3.0 (−1.1, −4.9) **	−0.79 (0.87, −2.4)	−1.87 (−0.06, −3.7) *	−14.5 (−1.6, −17.5) ***
**Children**	−4.3 (1.0, −9.5)	−3.9 (−2.6, 3.1)	5.1 (−1.6, 5.6)	−1.4 (−3.9, 0.99)	−5.1 (−0.69, −9.5) **
**General/high**	−10.2 (−14.8, −5.6) ***	−2.5 (−5.3, 0.21)	−0.64 (−2.8, 1.6)	−2.1 (−4.4, 0.11)	−15.5 (−19.8, −11.2)***
**General/medium−high**	−8.0 (−12.2, −3.8) **	−5.8 (−10.2, −1.3) **	−1.1 (−4.1, 1.9)	−0.72 (−4.1, 2.6)	−15.6 (−21.4, −9.7) ***
**General/medium**	−12.4 (−20.3, −4.5) **	−1.2 (−6.1, 3.7)	0.53 (−6.1, 7.2)	−7.1 (−14.8, 0.62)	−19.0 (−28.6, −9.5) **

* Data presented as APC/year mean diff. (%) and 95%CI; ** *p* < 0.03–0.01; *** *p* < 0001–0.00001 annual variability in APC of CT rate from 2014 to 2019.

## Data Availability

Not applicable.

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
