# Peer review of "Computed Tomography Utilization in the Management of Children with Mild Head Trauma"

_children, 2023, doi:10.3390/children10071274_

Round 1

Reviewer 1 Report

The manuscript presented from Leva et al., is interesting and well written, However the authors should improve the introduction comparing with previous studies in literature. Next, it is not clear if this study received ethical approval, The authors should explain it. 

Author Response

Response to Reviewer 1 Comments

Point 1: The manuscript presented by Leva et al., is interesting and well-written, However, the authors should improve the introduction comparing with previous studies in the literature. Next, it is not clear if this study received ethical approval, The authors should explain it. 

Response 1: As we mentioned in Introduction, “To our knowledge, no study has evaluated the effort of professional organizations in promoting PECARN rules in the hospital settings to reduce the use of CT in managing pediatric patients with mTBI.” We also explain the significant reduction of CT use after implementation of PECARN in children presenting with mTBI in single ED settings [19, 20, 21, 22]. We compared our findings with relevant publications in the Discussion [Please see Lines 226 to 256].  

Our study was approved by the institutional review board (IRB). The statement The study was conducted according to the guidelines of the Declaration of Helsinki and approved by the Rutgers Robert Wood Johnson Medical School Institutional Review Board (protocol code 2020000165, date of approval 3/5/2020)” is shown in the manuscript after the Conclusion.

Reviewer 2 Report

The authors present a manuscript describing trends in CT usage for children with mTBI following the initiation of the Safe CT Imaging Collaborative Initiative in acute care hospitals in the state of New Jersey. Overall, it is well conceived and written and will contribute to the field of pediatric TBI. A few suggestions for edits: 

Introduction:

-page 1 line 48-The term "malignancy" does not see applicable to pediatric TBI. Best so say - effects of TBI or something like that.

Methods:

-page 3 line 107- did you also include the category "head injury unspecified"?

Results:

-page 4 line 167- more seen in general hospital- can you please explain?

-page 4 line 183-what does this mean-reduction in 2016? Please add further details.

-page 4 line 187-Please clarify this sentence. It is hard to follow.

Discussion:

-page 6 line 252-"parental imaging expectations"- what does this mean- that parents requested imaging?

No comments.

Author Response

Response to Reviewer 2 Comments

The authors present a manuscript describing trends in CT usage for children with mTBI following the initiation of the Safe CT Imaging Collaborative Initiative in acute care hospitals in the state of New Jersey. Overall, it is well-conceived and written and will contribute to the field of pediatric TBI. A few suggestions for edits: 

Point 1: Introduction:

-page 1 line 48-The term "malignancy" does not seem applicable to pediatric TBI. Best so say - effects of TBI or something like that.

Response 1: Thank you for your comment. The sentence has been re-organized as follows: “In addition to the risk of death and severe disability, exposure to cranial computed tomography (CT), which is a primary diagnostic tool for managing TBI in childhood, could result in a lifelong threat of malignancy [2, 3, 4].

Point 2. Methods:

-page 3 line 107- did you also include the category "head injury unspecified"?

Response 2: The ICD-9 code 854 defines “Intracranial injury of other and unspecified nature” and ICD-10 code S09. 90 illustrates “Unspecified injury of the head.” The text “head injury without further description” covers the head injury unspecified. 

Point 3: Results

-page 4 line 167- more seen in a general hospital- can you please explain?

-page 4 line 183-what does this mean-reduction in 2016? Please add further details.

-page 4 line 187-Please clarify this sentence. It is hard to follow.

Response 3 (Page 4 line 167): Among all children seen in children's hospitals, the proportion of pediatric patients with mTBI was less than those in the general hospitals [2.7% (95%CI 2.3%, 3.1%) vs. 3.3% (95%CI 3.1%, 3.4%)], P<0.02.

Response 3 (Page 4 line 183): 2016 was the year of implementation of the Safe CT Imaging Collaborative Initiative. The correction has been made as follows: “CT use was significantly reduced in 2016, the year of implementation of Safe CT Imaging Collaborative Initiative, for almost all categories of patients and in 2017 for patients aged 0-1 and 15-17 years and Asian and other races.”

Response 3 (Page 4 line 187): The sentence “The overall baseline CT rate was 34.7% (95%CI 31.1%, 38.3%)” describes the average CT rate with a 95% Confidence Interval during the baseline period (2014-2015) before the implementation of the Safe CT Imaging Collaborative Initiative.

The correction has been made as follows: “Before the implementation of the Safe CT Imaging Collaborative Initiative (2014-2015), CT was used in the management of 34.7% (95%CI 31.1%, 38.3%) of children with mTBI.”

Point 4: Discussion:

-page 6 line 252-"parental imaging expectations"- what does this mean - that the parents requested imaging?

Response 4: The reference study [20] considered parental imaging expectations as a barrier to the implementation of PECARN rules in the management of children receiving CT. Management of parental expectations included ED providers meeting with the parents to show them the PECARN pathway and to explain the risk. However, the study did not analyze parental expectations regarding the use of CT for managing their children.